# Monash Time Series Forecasting Archive

**Rakshitha Godahewa**
Monash University
Melbourne, Australia
rakshitha.godahewa@monash.edu

**Christoph Bergmeir**
Monash University
Melbourne, Australia
christoph.bergmeir@monash.edu

**Geoffrey I. Webb**
Monash University
Melbourne, Australia
geoff.webb@monash.edu

**Rob J. Hyndman**
Monash University
Melbourne, Australia
rob.hyndman@monash.edu

**Pablo Montero-Manso**
University of Sydney
Australia
pmontm@gmail.com

## Abstract

Many businesses nowadays rely on large quantities of time series data making time series forecasting an important research area. Global forecasting models and multivariate models that are trained across sets of time series have shown huge potential in providing accurate forecasts compared with the traditional univariate forecasting models that work on isolated series. However, there are currently no comprehensive time series forecasting archives that contain datasets of time series from similar sources available for researchers to evaluate the performance of new global or multivariate forecasting algorithms over varied datasets. In this paper, we present such a comprehensive forecasting archive containing 25 publicly available time series datasets from varied domains, with different characteristics in terms of frequency, series lengths, and inclusion of missing values. We also characterise the datasets, and identify similarities and differences among them, by conducting a feature analysis. Furthermore, we present the performance of a set of standard baseline forecasting methods over all datasets across ten error metrics, for the benefit of researchers using the archive to benchmark their forecasting algorithms.

## 1 Introduction

Accurate time series forecasting is important for many businesses and industries to make decisions, and consequently, time series forecasting is a popular research area. The field of forecasting has traditionally been advanced by influential forecasting competitions. The most popular forecasting competition series is the M-competition series [1–5]. Other well-known forecasting competitions include the NN3 and NN5 Neural Network competitions [6], and Kaggle competitions such as the Wikipedia web traffic competition [7].

The winning approaches of many of the most recent competitions such as the winning method of the M4 by Smyl [8] and the winning method of the M5 forecasting competition [5], consist of global forecasting models [9] which train a single model across all series that need to be forecast. Compared with local univariate models, global forecasting models have the ability to learn cross-series information during model training and can control model complexity and overfitting on a global level [10]. This can be seen as a paradigm shift in forecasting. Over decades, single time series were seen as a dataset that should be studied and modelled in isolation. Nowadays, we are oftentimes interested in models built on sets of series from similar sources, such as series which are all product sales from a particular store, or series which are all smart meter readings in a particular city. Here, time series are seen as an instance in a dataset of many time series, to be studied and modelled together.

35th Conference on Neural Information Processing Systems (NeurIPS 2021) Track on Datasets and Benchmarks.

Closely related is the concept of multivariate forecasting, where multivariate models train and test over time series that are all the same length and all aligned in time, so that dependencies between series can be modelled. However, the concepts of global and multivariate models are in fact orthogonal, with local-univariate models being models for single time series, global-univariate models being models that train across series, but predict each series in isolation, with no need for the time series to have the same length or to be aligned in time, local-multivariate models modelling series that are all aligned in time and predicted together, and finally, global-multivariate models building one global model across (un-aligned) multivariate series. For the sake of simplicity, in the rest of the paper, we name local-univariate models as local or univariate models, global-univariate models as global models, local-multivariate models as multivariate models, and we deem global-multivariate models outside the scope of the paper. We further note that for each dataset for which multivariate methods are applicable, also global methods are applicable, but not vice versa. Thus, global models are more general than multivariate models.

Both global and multivariate models get attention lately in machine learning (especially deep learning), with Li et al. [11], Rangapuram et al. [12], Wen et al. [13] presenting global models and Salinas et al. [14], Sen et al. [15], Yu et al. [16], Zhou et al. [17] discussing novel approaches for multivariate modelling. However, when it comes to benchmarking, these recent works use a mere two [13] to seven [11] datasets to evaluate the performance of the new algorithms and the chosen datasets are different in each work. The datasets mainly belong to the energy, transport, and sales domains, and they do not include datasets from other domains such as banking, healthcare, or environmental applications.

In contrast, other areas of machine learning, such as general classification and regression, or time series classification, have greatly benefitted from established benchmark dataset archives, which allow a much broader and more standardised evaluation. The University of California Irvine (UCI) repository [18] is the most common and well-known benchmarking archive used in general machine learning, with currently 507 datasets from various domains. In time series classification, the dataset archives from the University of California Riverside (UCR) [19] and from the University of East Anglia (UEA) [20], contain 128 sets of univariate time series, and 30 datasets with multivariate time series, respectively, allowing routinely for much broader and more standardised evaluations of the methods, and therewith enabling more streamlined, robust, and reliable progress in the field.

The time series classification datasets, though they contain time series, do usually not resemble meaningful forecasting problems, so they cannot be used for the evaluation of forecasting methods. Also in the time series forecasting space there are a number of benchmarking archives, but they follow the paradigm of single series as datasets, and consequently contain mostly unrelated single time series. Examples are the Time Series Data Library [21], ForeDeCk [22], and Libra [23].

To the best of our knowledge, there exist currently no comprehensive time series forecasting benchmarking archives that focus on sets of time series to evaluate the performance of global and multivariate forecasting algorithms. We introduce such an archive, available at `https://forecastingdata.org/`. The archive contains 25 datasets including both previously publicly available time series datasets converted by us into a uniform format and made available at a central repository, as well as datasets curated by us. The datasets cover varied domains, with both equal and variable lengths time series. Many datasets have different versions based on the frequency and the inclusion of missing values, resulting in a total of 58 dataset variations.

We also introduce a new format to store time series data, based on the Weka ARFF file format [24], to overcome some of the shortcomings we observe in the .ts format used in the sktime time series repository [25]. We use a .tsf extension for this new format. This format stores the meta-information about a particular time series dataset such as dataset name, frequency, and inclusion of missing values as well as series specific information such as starting timestamps, in a non-redundant way. The format is very flexible and capable of including any other attributes related to time series as preferred by the users.

Furthermore, we analyse the characteristics of different series to identify the similarities and differences among them. For that, we conduct a feature analysis using tsfeatures [26] and catch22 features [27] extracted from all series of all datasets. The extracted features are publicly available for further research use. The performance of a set of baseline forecasting models including both traditional univariate forecasting models and global forecasting models are also evaluated over all datasets across ten error metrics. The forecasts and evaluation results of the baseline methods are

publicly available for the benefits of researchers that use the repository to benchmark their forecasting algorithms. All implementations to replicate the benchmark results and feature extraction, as well as code for loading the datasets into the R and Python environments, are publicly available at `https://github.com/rakshitha123/TSForecasting` .

## 2 Data records

This section details the datasets in our time series forecasting archive. The current archive contains 25 time series datasets. Furthermore, the archive contains in addition 5 single very long time series. As a large amount of data oftentimes renders machine learning methods feasible compared with traditional statistical modelling, and we are not aware of good and systematic benchmark data in this space either, these series are included in our repository as well. The datasets have different sampling rates such as yearly, quarterly, and monthly, and even high-frequency data with the highest sampling rate in the repository being a 4-secondly dataset. A summary of all primary datasets included in the repository is shown in Table 1. The table also reports whether a dataset is multivariate or not, meaning that it is aligned in time with known time stamps, and thus multivariate methods are applicable to the dataset. Global and univariate methods are applicable to all datasets in the repository.

A total of 58 datasets have been derived from these 30 primary datasets. Nine datasets contain time series belonging to different frequencies and the archive contains a separate dataset per each frequency. Eleven of the datasets have series with missing values. The archive contains 2 versions of each of these, one with and one without missing values. In the latter case, the missing values have been replaced by using an appropriate imputation technique.

Table 1: Datasets in the current time series forecasting archive

|   | Dataset | Domain | No: of Series | Min. Length | Max. Length | No: of Freq. | Missing | Competition | Multi-variate |
|---|---------|--------|---------------|-------------|-------------|--------------|---------|-------------|---------------|
| 1 | M1 | Multiple | 1001 | 15 | 150 | 3 | No | Yes | No |
| 2 | M3 | Multiple | 3003 | 20 | 144 | 4 | No | Yes | No |
| 3 | M4 | Multiple | 100000 | 19 | 9933 | 6 | No | Yes | No |
| 4 | Tourism | Tourism | 1311 | 11 | 333 | 3 | No | Yes | No |
| 5 | CIF 2016 | Banking | 72 | 34 | 120 | 1 | No | Yes | No |
| 6 | London Smart Meters | Energy | 5560 | 288 | 39648 | 1 | Yes | No | No |
| 7 | Aus. Electricity Demand | Energy | 5 | 230736 | 232272 | 1 | No | No | No |
| 8 | Wind Farms | Energy | 339 | 6345 | 527040 | 1 | Yes | No | No |
| 9 | Dominick | Sales | 115704 | 28 | 393 | 1 | No | No | No |
| 10 | Bitcoin | Economic | 18 | 2659 | 4581 | 1 | Yes | No | No |
| 11 | Pedestrian Counts | Transport | 66 | 576 | 96424 | 1 | No | No | No |
| 12 | Vehicle Trips | Transport | 329 | 70 | 243 | 1 | Yes | No | No |
| 13 | KDD Cup 2018 | Nature | 270 | 9504 | 10920 | 1 | Yes | Yes | No |
| 14 | Weather | Nature | 3010 | 1332 | 65981 | 1 | No | No | No |
| 15 | NN5 | Banking | 111 | 791 | 791 | 2 | Yes | Yes | Yes |
| 16 | Web Traffic | Web | 145063 | 803 | 803 | 2 | Yes | Yes | Yes |
| 17 | Solar | Energy | 137 | 52560 | 52560 | 2 | No | No | Yes |
| 18 | Electricity | Energy | 321 | 26304 | 26304 | 2 | No | No | Yes |
| 19 | Car Parts | Sales | 2674 | 51 | 51 | 1 | Yes | No | Yes |
| 20 | FRED-MD | Economic | 107 | 728 | 728 | 1 | No | No | Yes |
| 21 | San Francisco Traffic | Transport | 862 | 17544 | 17544 | 2 | No | No | Yes |
| 22 | Rideshare | Transport | 2304 | 541 | 541 | 1 | Yes | No | Yes |
| 23 | Hospital | Health | 767 | 84 | 84 | 1 | No | No | Yes |
| 24 | COVID Deaths | Nature | 266 | 212 | 212 | 1 | No | No | Yes |
| 25 | Temperature Rain | Nature | 32072 | 725 | 725 | 1 | Yes | No | Yes |
| 26 | Sunspot | Nature | 1 | 73931 | 73931 | 1 | Yes | No | No |
| 27 | Saugeen River Flow | Nature | 1 | 23741 | 23741 | 1 | No | No | No |
| 28 | US Births | Nature | 1 | 7305 | 7305 | 1 | No | No | No |
| 29 | Solar Power | Energy | 1 | 7397222 | 7397222 | 1 | No | No | No |
| 30 | Wind Power | Energy | 1 | 7397147 | 7397147 | 1 | No | No | No |

Out of the 30 datasets, 7 have not been previously available in the current form, and we have undertaken significant work to curate them, namely: Australian Electricity Demand, Wind Farms, Bitcoin, Rideshare, Temperature Rain, Solar Power, and Wind Power. The remaining 23 datasets have been publicly available before in different formats, and we have converted them into a unified format and make them available at a unified source repository. From those, 8 originate from competition platforms, 3 from research conducted by Lai et al. [28], 5 are taken from R packages, 1 is from

the Kaggle platform [29], and 1 is taken from a Johns Hopkins repository [30]. The other datasets have been extracted from corresponding domain specific platforms. The datasets mainly belong to 9 different domains: tourism, banking, web, energy, sales, economics, transportation, health, and nature. Three datasets, the M1 [1], M3 [2], and M4 [3, 4] datasets, contain series belonging to multiple domains. We furthermore ensured that all datasets have licenses that allow us to include them in the repository. All datasets and the corresponding data collection procedures are explained in detail in the Appendix (supplementary materials).

## 2.1 Data format

We introduce a new format to store time series data, based on the Weka ARFF file format [24]. We use a .tsf file extension. Our format is comparable with the .ts format used in the sktime repository [25], but we deem it more streamlined and more flexible. The basic idea of the file format is that each data file can contain 1) attributes that are constant throughout the whole dataset (e.g., the forecasting horizon, whether the dataset contains missing values or not), 2) attributes that are constant throughout a time series (e.g., its name, its position in a hierarchy, product information for product sales time series), and 3) attributes that are particular to each data point (the value of the series, or timestamps for non-equally spaced series). An example of series in this format is shown in the Appendix (supplementary materials).

Each .tsf file contains tags describing the meta-information of the corresponding dataset such as *@frequency* (seasonality), *@horizon* (expected forecast horizon), *@missing* (whether the series contain missing values) and *@equallength* (whether the series have equal lengths). We note that in principle these attributes can be freely defined by the user and the file format does not need any of these values to be defined in a certain way, though the file readers reading the files may rely on existence of attributes with certain names and assume certain meanings. Next, there are attributes in each dataset which describe series-wise properties, where the tag *@attribute* is followed by the name and type. Examples are *series_name* (the unique identifier of a given series) and *start_ timestamp* (the start timestamp of a given series). Again, the format has the flexibility to include any additional series-wise attributes as preferred by the users.

Following the ARFF file format, the data are then listed under the *@data* tag after defining attributes and meta-headers, and attribute values are separated by colons. The only extension that our format has, compared with the original ARFF file format, is that the time series then are appended to their attribute vector as a comma-separated variable-length vector. As this vector can have a different length for each instance, this cannot be represented in the original ARFF file format. In particular, a time series with $m$ number of attributes and $n$ number of values can be represented as:

$$< attribute_1 >:< attribute_2 >: ... :< attribute_m >:< s_1, s_2, ..., s_n > \qquad (1)$$

The missing values in the series are indicated using the "?" symbol. Code to load datasets in this format into R and Python is available in our github repository at `https://github.com/rakshitha123/TSForecasting`.

## 3 Feature Analysis

This section details the feature analysis we conducted on the datasets in our repository and its results.

### 3.1 Feature analysis methodology

We characterise the datasets in our archive to analyse the similarities and differences between them, to gain a better understanding on where gaps in the repository may be and what type of data are prevalent in applications. This may also help to select suitable forecasting methods for different types of datasets. We analyse the characteristics of the datasets using the tsfeatures [26] and catch22 [27] feature extraction methods. All extracted features are publicly available on our website `https://forecastingdata.org/` for further research use. Due to the large size of the datasets, we have not been able to extract features from the London smart meters, wind farms, solar power, and wind power datasets, which is why we exclude them from this analysis.

We extract 42 features using the *tsfeatures* function in the R package *tsfeatures* [26] including mean, variance, autocorrelation features, seasonal features, entropy, crossing points,

flat spots, lumpiness, non-linearity, stability, Holt-parameters, and features related to the Kwiatkowski–Phillips–Schmidt–Shin (KPSS) test [31] and the Phillips–Perron (PP) test [32]. For all series that have a frequency greater than daily, we consider multi-seasonal frequencies when computing features. Therefore, the number of features extracted is higher for multi-seasonal datasets as the seasonal features are individually calculated for each season presented in the series. Furthermore, if a series is short and does not contain two full seasonal cycles, we calculate the features assuming a non-seasonal series (i.e., setting its frequency to "one" for the feature extraction). We use the *catch22_all* function in the R package *Rcatch22* [33] to extract the catch22 features from a given time series. The features are a subset of 22 features from the *hctsa* package [34] which includes the implementations of over 7000 time series features. The computational cost of the catch22 features is low compared with computing all features implemented in the hctsa package.

For the feature analysis, we consider 5 features, as suggested by Bojer and Meldgaard [35]: first order autocorrelation (ACF1), trend, entropy, seasonal strength, and the Box-Cox transformation parameter, lambda. The *BoxCox.lambda* function in the R package *forecast* [36] is used to extract the Box-Cox transformation parameter from each series, with default parameters. The other 4 features are extracted using *tsfeatures*. Since this feature space contains 5 dimensions, to compare and visualise the features across multiple datasets, we reduce the feature dimensionality to 2 using Principal Component Analysis [PCA, 37].

The numbers of series in each dataset are significantly different, e.g., the CIF 2016 monthly dataset and M4 monthly dataset contain 72 and 48,000 series, respectively. Hence, if all series were considered to calculate the PCA components, those components would be dominated by datasets that have large amounts of series. Therefore, for datasets that contain more than 300 series, we randomly take a sample of 300 series, before constructing the PCA components across all datasets. Once the components are calculated, we map all series of all datasets into the resulting PCA feature space. We note that we use PCA for dimensionality reduction over other advanced dimensionality reduction algorithms such as t-Distributed Stochastic Neighbor Embedding [t-SNE, 38] due to this capability of constructing the basis of the feature space with a reduced sample of series with the possibility to then map all series into the space afterwards.

## 3.2   Feature analysis results

Figure 1 shows hexbin plots of the normalised density values of the low-dimensional feature space generated by PCA across ACF1, trend, entropy, seasonal strength and Box-Cox lambda for 20 selected datasets (plots for all datasets are available in the Appendix in the supplementary material). The figure highlights the characteristics among different datasets. For the M competition datasets, the feature space is highly populated on the left-hand side and hence, denoting high trend and ACF1 levels in the series. The tourism yearly dataset also shows high trend and ACF1 levels. In contrast, the car parts, hospital, and Kaggle web traffic datasets show high density levels towards the right-hand side, indicating a higher degree of entropy. The presence of intermittent series can be considered as the major reason for the higher degree of entropy in the Kaggle web traffic and car parts datasets. The plots confirm the claims of prior similar studies [35, 39] that the M competition datasets are significantly different from the Kaggle web traffic dataset.

The monthly datasets generally show high seasonal strengths compared with datasets of other frequencies. Quarterly datasets also demonstrate high seasonal strengths except for the M4 quarterly dataset. In contrast, the datasets with high frequencies such as weekly, daily, and hourly show low seasonal strengths except for the NN5 weekly and NN5 daily datasets.

Related to the shapes of the feature space, the 3 yearly datasets: M3, M4, and tourism show very similar shapes and density populations indicating they have similar characteristics. The M4 quarterly dataset also shows a similar shape as the yearly datasets, even though it has a different frequency. The other 2 quarterly datasets M3 and tourism are different, but similar to each other. The M3 and M4 monthly datasets are similar to each other in terms of both shape and density population. Furthermore, the electricity hourly and traffic hourly datasets have similar shapes and density populations, whereas the M4 hourly dataset has a slightly different shape compared with them. The daily datasets show different shapes and density populations, where the NN5 daily dataset is considerably different from the other 2 daily datasets: M4 and Kaggle web traffic, in terms of shape and all 3 daily datasets are considerably different from each other in terms of density population. The weekly datasets also show different shapes and density populations compared with each other.

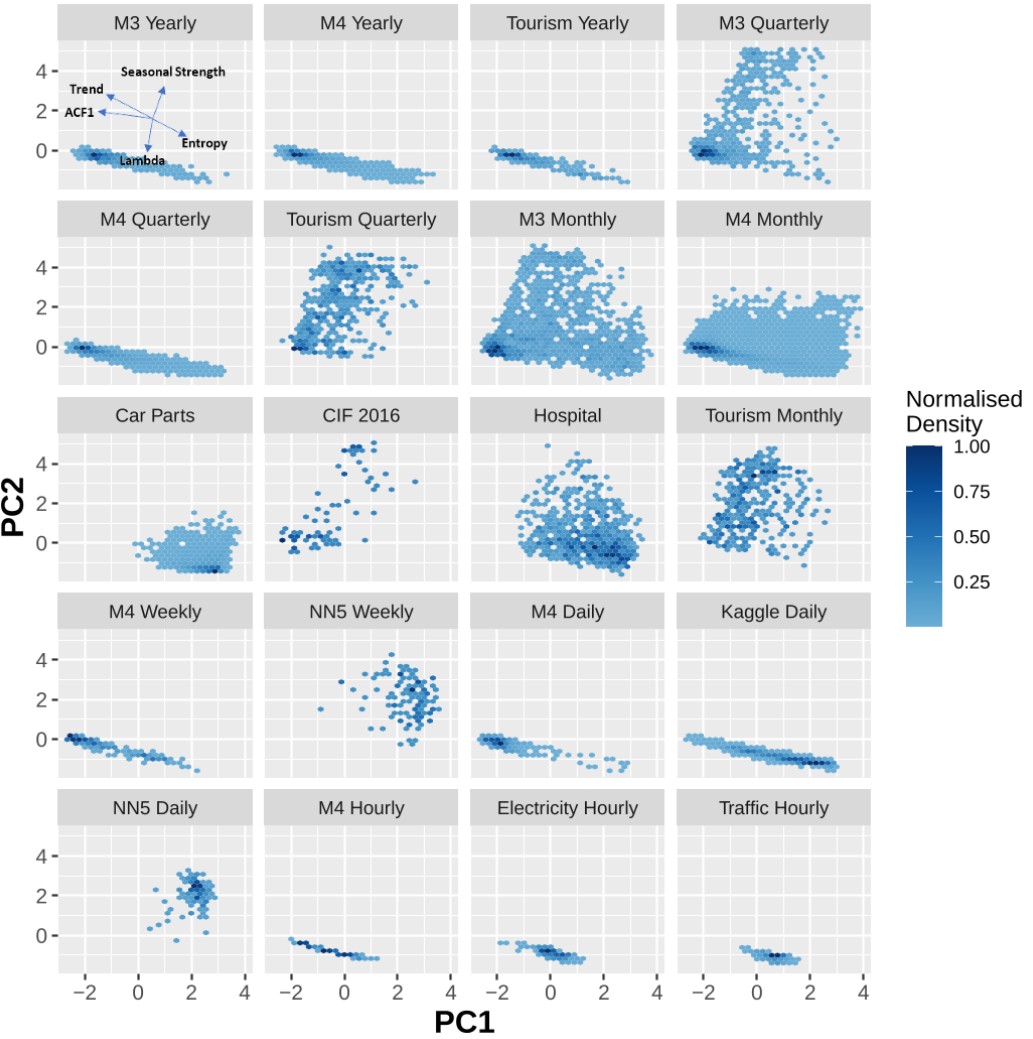

Figure 1: Hexbin plots showing the normalised density values of the low-dimensional feature space generated by PCA across ACF1, trend, entropy, seasonal strength, and Box-Cox lambda for 20 datasets. The dark and light hexbins denote the high and low density areas, respectively. The M3 Yearly facet shows the directions of the 5 features, which are the same across all facets.

## 4 Baseline evaluation

This section details the baseline evaluation we conducted on the datasets in our repository together with a discussion of the results.

### 4.1 Baseline evaluation methodology

In the forecasting space, benchmarking against simple benchmarks is vital [40] as even simple benchmarks can oftentimes be surprisingly competitive. However, many works in the machine learning space are notoriously weak when it comes to proper benchmarking for time series forecasting [41]. To fill this gap, we evaluate the performance of 13 different baseline forecasting models over the datasets in our repository using a fixed origin evaluation scheme, so that researchers that use the data in our repository can directly benchmark their forecasting algorithms against these baselines. The baseline models include 6 traditional univariate forecasting models: Exponential Smoothing [ETS, 42], Auto-Regressive Integrated Moving Average [ARIMA, 43], Simple Exponential Smoothing

(SES), Theta [44], Trigonometric Box-Cox ARMA Trend Seasonal [TBATS, 45] and Dynamic Harmonic Regression ARIMA [DHR-ARIMA, 46], and 7 global forecasting models: a linear Pooled Regression model [PR, 47], a Feed-Forward Neural Network [FFNN, 48], CatBoost [49], DeepAR [50], N-BEATS [51], a WaveNet [52], and a Transformer [53] method, covering a representative set of state-of-the-art forecasting models from statistical, machine learning, and deep learning domains.

We use the R packages *forecast* [54], *glmnet* [55], and *catboost* [49] to implement the 6 traditional univariate forecasting methods, the globally trained PR model, and CatBoost, respectively. For the remaining global models, we use the implementations of FFNN, DeepAR, N-BEATS, WaveNet, and Transformer available from the Python package *GluonTS* [56]. All models are executed with their default parameters since we present them as benchmarks in our study and note that they could still perform better on the datasets with additional hyperparameter tuning and deeper understanding of the methods. Furthermore, we note that traditionally forecasters have focused on the R programming language, and many of the statistical benchmarks from the *forecast* package have no direct correspondence in the Python programming language, which may have been a contributing factor in the past for weak forecasting evaluations. Thus, using the best implementations available across both Python and R and making the code available accordingly is a particular feature of our work.

Again, we do not consider the London smart meters, wind farms, solar power, and wind power datasets for both univariate and global model evaluations, the Kaggle web traffic daily dataset for the global model evaluations and the solar 10 minutely dataset for the WaveNet evaluation, as the computational cost of running these models was not feasible in our experimental environment. The M4 yearly dataset is not considered for the neural network benchmarks implemented using GluonTS since the corresponding implementations cannot handle the very long time series of the dataset spanning over 600 years.

Theta, SES, TBATS, ETS and the 7 global models are used for all datasets. ETS is used as a non-seasonal model for weekly and multi-seasonal datasets such as 10 minutely, half hourly, and hourly as the corresponding implementation cannot handle seasonal cycles greater than 24. We use 2 versions of ARIMA. The general ARIMA method is used for yearly, quarterly, monthly, and daily datasets whereas DHR-ARIMA is used for multi-seasonal datasets due to its capability of dealing with multiple seasonalities [57]. DHR-ARIMA is also used for weekly datasets due to its capability of dealing with long non-integer seasonal cycles present in weekly data [58].

Forecast horizons are chosen for each dataset to evaluate the model performance. For all competition datasets, we use the forecast horizons originally employed in the competitions. For the remaining datasets, 12 months ahead forecasts are obtained for monthly datasets, 8 weeks ahead forecasts are obtained for weekly datasets, except the solar weekly dataset, and 30 days ahead forecasts are obtained for daily datasets. For the solar weekly dataset, we use a horizon of 5 as the series in this dataset are relatively short compared with other weekly datasets. For half-hourly, hourly and other high-frequency datasets, we set the forecasting horizon to one week, e.g., 168 is used as the horizon for hourly datasets.

The number of lagged values used in the global models are determined based on a heuristic suggested in prior work [59]. Generally, the number of lagged values is chosen as the seasonality multiplied with 1.25. If the datasets contain short series and it is impossible to use the above defined number of lags, for example in the Dominick and solar weekly datasets, then the number of lagged values is chosen as the forecast horizon multiplied with 1.25, assuming that the horizon is not arbitrarily chosen but based on certain characteristics of the time series structure. When defining the number of lagged values for multi-seasonal datasets, we consider the corresponding weekly seasonality value, e.g., 168 for hourly datasets. If it is impossible to use the number of lagged values obtained with the weekly seasonality due to high memory and computational requirements, for example with the traffic hourly and electricity hourly datasets, then we use the corresponding daily seasonality value to define the number of lags, e.g., 24 for hourly datasets. In particular, due to high memory and computational requirements, the number of lagged values is chosen as 50 for the solar 10 minutely dataset which is less than the above mentioned heuristics based on seasonality and forecasting horizon suggest.

## 4.2 Baseline evaluation results

It is very difficult to define error measures for forecasting that perform well under all situations [46], in the sense that it is difficult to define a scale-free measure that works for any type of non-stationarity

in the time series. Furthermore, choosing suitable error metrics for a given forecasting task, is highly domain and application dependent. Thus, how to best evaluate forecasts is still an active area of research, and (especially in the machine learning area) researchers often use ad-hoc, non-adequate measures. For example, usage of the Mean Absolute Percentage Error (MAPE) for normalised data between 0 and 1 may result in undefined or heavily skewed measures, or error measures using the mean of a series like the Root Relative Squared Error [RSE, 28] will not work properly for series where the mean is essentially meaningless, such as series with steep trends. To address these issues, we analyse the performance of the baseline models over a broad range of error measures, so that characteristics of different error measures can be assessed across a broad range of datasets and forecasting methods, whereas the researchers can select a suitable error measure to benchmark their algorithms, depending on their domain and application. In particular, we use four error metrics that – while having their own problems – are common for evaluation in forecasting, namely the Mean Absolute Scaled Error [MASE, 60], symmetric MAPE (sMAPE), Mean Absolute Error [MAE, 61], and Root Mean Squared Error (RMSE) to evaluate the performance of the thirteen baseline forecasting models explained in Section 4.1. For datasets containing zeros, calculating the sMAPE error measure may lead to divisions by zero. Hence, we also consider the variant of the sMAPE proposed by Suilin [62] which overcomes the problems with small values and divisions by zero of the original sMAPE. We report the original sMAPE only for datasets where divisions by zero do not occur and the modified sMAPE (msMAPE) for all datasets. The definitions of all error metrics are available in the Appendix (supplementary materials).

As a general guideline, scale-dependent error measures such as MAE and RMSE should be considered first. They are easy to interpret and do not share many of the drawbacks of the other measures. MAE is minimal for a forecast that is the median of the forecasting distribution, while RMSE is minimal for its mean. As such, the only drawbacks are that the RMSE is more sensitive towards outliers and the optimal MAE for a series with over 50% zeros (intermittent time series) will lead to a forecast heavily biased towards zero. However, scale-dependent measures cannot be used to compare forecasts for series with significantly different scales, across or within datasets. A scale-free measure that works under any type of non-stationarity does not exist, to the best of our knowledge. Though the sMAPE is popular, the sMAPE (and related measures such as MAPE) is not a good error metric for datasets containing zero values (especially intermittent datasets) such as Kaggle web traffic and carparts. If there is a zero in the actual data and if a model does not predict an exact zero, then sMAPE takes its maximal value, 200, independent of the actual predicted value. Furthermore, sMAPE violates symmetry as under-prediction gives higher errors compared to over-prediction even though the absolute forecasting error is the same. However, a big advantage of the sMAPE is that it is bounded with a maximal value of 200. The msMAPE addresses some of these issues but at the cost of not having a theoretical underpinning as it does not optimise for a meaningful summary statistic of the forecasting distribution (such as the mean or median). We include sMAPE and msMAPE mostly for their popularity, to allow for easy comparisons. The MASE is arguably the closest measure to a generally applicable scale-free measure for time series. It avoids the issues stated for the other error metrics such as symmetry issues, scaling issues and division by zero; it is optimal for the median of the forecast distribution. Its only problems arise if the naive forecast has very different performance for different parts of a series (e.g., training and test sets), e.g., for very stable training periods that are followed by very volatile test periods. Thus, where a scale-free measure is needed, we use the MASE as the primary error metric of our study.

The MASE measures the performance of a model compared with the in-sample average performance of a one-step-ahead naïve or seasonal naïve (snaïve) benchmark. For multi-seasonal datasets, we use the length of the shortest seasonality to calculate the MASE. For the datasets where all series contain at least one full seasonal cycle of data points, we consider the series to be seasonal and calculate MASE values using the snaïve benchmark. Otherwise, we calculate the MASE using the naïve benchmark, effectively treating the series as non-seasonal.

The error metrics are defined for each series individually. We further calculate the mean and median values of the error metrics over the datasets to evaluate the model performance and hence, each model is evaluated using 10 error metrics for a particular dataset: mean MASE, median MASE, mean sMAPE, median sMAPE, mean msMAPE, median msMAPE, mean MAE, median MAE, mean RMSE and median RMSE. Table 2 shows the mean MASE of the thirteen baselines on all datasets. The results of all baselines across all datasets on all 10 error metrics are available in the Appendix.

Table 2: Mean MASE results. The best model across each dataset is highlighted in boldface.

| Dataset | SES | Theta | TBATS | ETS | (DHR-) ARIMA | PR | Cat Boost | FFNN | Deep AR | N-BEATS | Wave Net | Trans former |
|---|---|---|---|---|---|---|---|---|---|---|---|---|
| M1 Yearly | 4.938 | 4.191 | **3.499** | 3.771 | 4.479 | 4.588 | 4.427 | 4.355 | 4.603 | 4.384 | 4.666 | 5.519 |
| M1 Quarterly | 1.929 | 1.702 | 1.694 | **1.658** | 1.787 | 1.892 | 2.031 | 1.862 | 1.833 | 1.788 | 1.700 | 2.772 |
| M1 Monthly | 1.379 | 1.091 | 1.118 | **1.074** | 1.164 | 1.123 | 1.209 | 1.205 | 1.192 | 1.168 | 1.200 | 2.191 |
| M3 Yearly | 3.167 | **2.774** | 3.127 | 2.860 | 3.417 | 3.223 | 3.788 | 3.399 | 3.508 | 2.961 | 3.014 | 3.003 |
| M3 Quarterly | 1.417 | **1.117** | 1.256 | 1.170 | 1.240 | 1.248 | 1.441 | 1.329 | 1.310 | 1.182 | 1.290 | 2.452 |
| M3 Monthly | 1.091 | 0.864 | **0.861** | 0.865 | 0.873 | 1.010 | 1.065 | 1.011 | 1.167 | 0.934 | 1.008 | 1.454 |
| M3 Other | 3.089 | 2.271 | 1.848 | **1.814** | 1.831 | 2.655 | 3.178 | 2.615 | 2.975 | 2.390 | 2.127 | 2.781 |
| M4 Yearly | 3.981 | **3.375** | 3.437 | 3.444 | 3.876 | 3.625 | 3.649 | - | - | - | - | - |
| M4 Quarterly | 1.417 | 1.231 | 1.186 | **1.161** | 1.228 | 1.316 | 1.338 | 1.420 | 1.274 | 1.239 | 1.242 | 1.520 |
| M4 Monthly | 1.150 | 0.970 | 1.053 | **0.948** | 0.962 | 1.080 | 1.093 | 1.151 | 1.163 | 1.026 | 1.160 | 2.125 |
| M4 Weekly | 0.587 | 0.546 | 0.504 | 0.575 | 0.550 | 0.481 | 0.615 | 0.545 | 0.586 | **0.453** | 0.587 | 0.695 |
| M4 Daily | 1.154 | 1.153 | 1.157 | 1.239 | 1.179 | 1.162 | 1.593 | **1.141** | 2.212 | 1.218 | 1.157 | 1.377 |
| M4 Hourly | 11.607 | 11.524 | 2.663 | 26.690 | 13.557 | **1.662** | 1.771 | 2.862 | 2.145 | 2.247 | 1.680 | 8.840 |
| Tourism Yearly | 3.253 | 3.015 | 3.685 | 3.395 | 3.775 | 3.516 | 3.553 | 3.401 | 3.205 | **2.977** | 3.624 | 3.552 |
| Tourism Quarterly | 3.210 | 1.661 | 1.835 | 1.592 | 1.782 | 1.643 | 1.793 | 1.678 | 1.597 | **1.475** | 1.714 | 1.859 |
| Tourism Monthly | 3.306 | 1.649 | 1.751 | 1.526 | 1.589 | 1.678 | 1.699 | 1.582 | **1.409** | 1.574 | 1.482 | 1.571 |
| CIF 2016 | 1.291 | 0.997 | 0.861 | **0.841** | 0.929 | 1.019 | 1.175 | 1.053 | 1.159 | 0.971 | 1.800 | 1.173 |
| Aus. Elecdemand | 1.857 | 1.867 | 1.174 | 5.663 | 2.574 | 0.780 | **0.705** | 1.222 | 1.591 | 1.014 | 1.102 | 1.113 |
| Dominick | 0.582 | 0.610 | 0.722 | 0.595 | 0.796 | 0.980 | 1.038 | 0.614 | 0.540 | 0.952 | **0.531** | 0.531 |
| Bitcoin | 4.327 | 4.344 | 4.611 | 2.718 | 4.030 | **2.664** | 2.888 | 6.006 | 6.394 | 7.254 | 5.315 | 8.462 |
| Pedestrians | 0.957 | 0.958 | 1.297 | 1.190 | 3.947 | 0.256 | 0.262 | 0.267 | 0.272 | 0.380 | **0.247** | 0.274 |
| Vehicle Trips | 1.224 | 1.244 | 1.860 | 1.305 | 1.282 | 1.212 | **1.176** | 1.843 | 1.929 | 2.143 | 1.851 | 2.532 |
| KDD | 1.645 | 1.646 | 1.394 | 1.787 | 1.982 | 1.265 | 1.233 | 1.228 | 1.699 | 1.600 | **1.185** | 1.696 |
| Weather | 0.677 | 0.749 | 0.689 | 0.702 | 0.746 | 3.046 | 0.762 | 0.638 | **0.631** | 0.717 | 0.721 | 0.650 |
| NN5 Daily | 1.521 | 0.885 | **0.858** | 0.865 | 1.013 | 1.263 | 0.973 | 0.941 | 0.919 | 1.134 | 0.916 | 0.958 |
| NN5 Weekly | 0.903 | 0.885 | 0.872 | 0.911 | 0.887 | 0.854 | 0.853 | 0.850 | 0.863 | **0.808** | 1.123 | 1.141 |
| Kaggle Daily | 0.924 | 0.928 | 0.947 | 1.231 | **0.890** | - | - | - | - | - | - | - |
| Kaggle Weekly | 0.698 | 0.694 | **0.622** | 0.770 | 0.815 | 1.021 | 1.928 | 0.689 | 0.758 | 0.667 | 0.628 | 0.888 |
| Solar 10 Mins | 1.451 | 1.452 | 3.936 | 1.451 | **1.034** | 1.451 | 2.504 | 1.450 | 1.450 | 1.573 | - | 1.451 |
| Solar Weekly | 1.215 | 1.224 | 0.916 | 1.134 | 0.848 | 1.053 | 1.530 | 1.045 | 0.725 | 1.184 | 1.961 | **0.574** |
| Electricity Hourly | 4.544 | 4.545 | 3.690 | 6.501 | 4.602 | 2.912 | 2.262 | 3.200 | 2.516 | 1.968 | **1.606** | 2.522 |
| Electricity Weekly | 1.536 | 1.476 | 0.792 | 1.526 | 0.878 | 0.916 | 0.815 | **0.769** | 1.005 | 0.800 | 1.250 | 1.770 |
| Carparts | 0.897 | 0.914 | 0.998 | 0.925 | 0.926 | 0.755 | 0.853 | 0.747 | 0.747 | 2.836 | 0.754 | **0.746** |
| FRED-MD | 0.617 | 0.698 | 0.502 | **0.468** | 0.533 | 8.827 | 0.947 | 0.601 | 0.640 | 0.604 | 0.806 | 1.823 |
| Traffic Hourly | 1.922 | 1.922 | 2.482 | 2.294 | 2.535 | 1.281 | 1.571 | 0.892 | 0.825 | 1.100 | 1.066 | **0.821** |
| Traffic Weekly | 1.116 | 1.121 | 1.148 | 1.125 | 1.191 | 1.122 | 1.116 | 1.150 | 1.182 | **1.094** | 1.233 | 1.555 |
| Rideshare | 3.014 | 3.641 | 3.067 | 4.040 | **1.530** | 3.019 | 2.908 | 4.198 | 4.029 | 3.877 | 3.009 | 4.040 |
| Hospital | 0.813 | **0.761** | 0.768 | 0.765 | 0.787 | 0.782 | 0.798 | 0.840 | 0.769 | 0.791 | 0.779 | 1.031 |
| COVID | 7.776 | 7.793 | 5.719 | **5.326** | 6.117 | 8.731 | 8.241 | 5.459 | 6.895 | 5.858 | 7.835 | 8.941 |
| Temp. Rain | 1.347 | 1.368 | 1.227 | 1.401 | 1.174 | 0.876 | 1.028 | 0.847 | 0.785 | 1.300 | 0.786 | **0.687** |
| Sunspot | 0.128 | 0.128 | 0.067 | 0.128 | 0.067 | 0.099 | 0.059 | 0.207 | 0.020 | 0.375 | 0.004 | **0.003** |
| Saugeen | 1.426 | 1.425 | 1.477 | 2.036 | 1.485 | 1.674 | **1.411** | 1.524 | 1.560 | 1.852 | 1.471 | 1.861 |
| Births | 4.343 | 2.138 | **1.453** | 1.529 | 1.917 | 2.094 | 1.609 | 2.032 | 1.548 | 1.537 | 1.837 | 1.650 |

Overall, SES shows the worst performance and Theta shows the second-worst performance across all error metrics. TBATS, ETS and ARIMA show a mixed performance on the yearly, monthly, quarterly, and daily datasets but all outperform SES and Theta. TBATS generally shows a better performance than DHR-ARIMA on the high frequency datasets. For our experiments, we always set the maximum order of Fourier terms used with DHR-ARIMA to $k = 1$. Based on the characteristics of the datasets, $k$ can be tuned as a hyperparameter and it may lead to better results compared with our results. Compared with SES and Theta, DHR-ARIMA shows a superior performance.

The global models show a mixed performance compared with the traditional univariate forecasting models. Overall, CatBoost and FFNN show the worst performance whereas the PR models show a mixed performance. The deep learning models such as DeepAR, N-BEATS, WaveNet and Transformer models show a better performance compared to the other considered global benchmarks. The performance of the global models is considerably affected by the number of lags used during model training, performing better as the number of lags is increased. The number of lags we use during model training is quite high with the high-frequency datasets such as hourly, compared with the other

datasets and hence, global models generally show a better performance than the traditional univariate forecasting models on all error metrics across those datasets. But on the other hand, the memory and computational requirements are also increased when training global models with larger numbers of lags. Furthermore, the global models show better performance across intermittent datasets such as car parts, compared with the traditional univariate forecasting models. The error measures are not directly comparable across datasets as we consider different forecasting horizons for varied datasets.

We see that the simple univariate forecasting models can provide better results for datasets with short, unrelated, noisy series, such as many of the M-competition datasets, where in fact often the simple Theta method still outperforms all more sophisticated methods. For the intermittent datasets such as carparts, the performance of the included simple univariate benchmark models are relatively worse, whereas global machine learning or deep learning models seem to be able to better handle the intermittency without special adjustments. Relating these results to our feature analysis, in general, we see the simple univariate benchmarks can properly learn the patterns in datasets with a higher degree of trend and ACF1, whereas machine learning and deep learning models are generally better at forecasting the datasets with a higher degree of entropy or uncertainty.

The execution times of the baselines across all datasets are available in the Appendix. The deep learning models show competitive execution times compared to other baselines.

## 5 Conclusions and future research

Recently, global forecasting models and multivariate models have shown huge potential in providing accurate forecasts for collections of time series compared with the traditional univariate benchmarks. However, there are currently no comprehensive time series forecasting benchmark data archives available that contain datasets to facilitate the evaluation of these new forecasting algorithms. In this paper, we have presented the details of an archive that contains 25 publicly available time series datasets with different frequencies from varied domains. We have also characterised the datasets and have identified the similarities and differences among them by conducting a feature analysis exercise using tsfeatures and catch22 features extracted from each series. Finally, we have evaluated the performance of thirteen baseline forecasting models over all datasets across ten error metrics to enable other researchers to benchmark their own forecasting algorithms directly against those.

Some areas for further research are as follows. In this research, we have only used univariate forecasting benchmarks. Analysing the performance of multivariate forecasting benchmarks is also an important possible future work. Furthermore, we have only considered point forecasts where probabilistic forecasting performance of the benchmarks is also interesting to study.

We also expect to run new baseline models in the future and the results tables in our website will be updated accordingly. The users are also invited to integrate new models to our framework based on the instructions given in our code repository, and to send us the results of the new forecasting models. If computationally feasible, we expect to re-execute the models and confirm the results. In the future, we expect to maintain two results tables in our website with the confirmed and unconfirmed results of the forecasting models.

## Acknowledgements

This research was supported by the Australian Research Council under grant DE190100045, a Facebook Statistics for Improving Insights and Decisions research award, Monash University Graduate Research funding and the MASSIVE High performance computing facility, Australia.

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
