# OpenReview forum: "Monash Time Series Forecasting Archive"
_NeurIPS.cc/2021/Track/Datasets_and_Benchmarks/Round2 — NeurIPS 2021 Datasets and Benchmarks Track (Round 2)_

### Official Review · Reviewer_X5f9 · 2021-09-19
**Strong datasets but unfair benchmarking due to lack of preprocessing for neural models**

**Rating:** 6
**Confidence:** 4
**Clarity:** Yes

**Strengths:**

- The paper is well written and clear.
- This benchmark has huge potential in advancing state of the art forecasting methods (if processed correctly).
- The code is clear and can be easily reproduced.
- Looking at the previous reviews in the supplementary material provided by the authors; it seems like most issues have been addressed in the new draft.

**Weaknesses:**

- The paper shows that statical methods tend to outperform neural networks in most datasets which is quite surprising since given enough capacity and data a neural network should be able to perform as well as simple statical methods. Since this was quite surprising for me I looked into the code to reproduce the results and one thing I noticed is that data is downloaded from the archive and passed to the networks. I believe the data is in its raw form (please correct me if I am wrong) so no data preprocessing is done whatsoever before feeding it into the networks which might be the reason behind such results.    Preprocessing timeseries is tricky especially that different datasets might need different processing steps.

A complete forecasting benchmark would involve the following:
Download data -> Process data -> Feed to models -> evaluate models.

Currently, step two is missing which might lead to undesirable behavior, especially for neural networks.

**Additional Feedback:**

- Please clarify, is the data on the archive saved as the raw data or preprocessed?
- Are all the results for univariate forecasting? using only a single variable or all available features are used as inputs? Since it seems that neural network only get univariate features (take covid death as an example)

########Post rebuttal

Thank you for the clarification.
Given that preprocessing is done within the GluonTS, I raise my score to 6.

**Correctness:**

Currently, the benchmark is missing preprocessing data phase which might lead to incorrect results when comparing different machine learning methods

**Documentation:**

Yes

**Relation To Prior Work:**

Yes

**Summary And Contributions:**

The paper introduces an archive consisting of a total of 30 time series datasets. 7 of the dataset are newly introduced in this paper remaining datasets have already been publically available. The paper also introduces TSF formate which is a new formate to save time series data. The paper benchmarked different machine learning methods both traditional methods and neural methods.

---

> ### Author Response · Authors · 2021-09-26
> **Responses to the Reviewer X5f9 Comments**
>
> We thank the reviewer for recognising the potential of our work.
>
> Regarding preprocessing for Neural Networks (NN), as the reviewer states, “preprocessing timeseries is tricky especially that different datasets might need different processing steps”. We would argue that in fact it is still an active research area [C1]. As such, we consider the preprocessing as part of the method. In particular, all 5 NN models that we have used in the paper are implemented using the popular GluonTS framework. This framework internally handles data preprocessing. It expects the users to provide raw data and time stamps as input for model training, and performs scaling and seasonality handling (e.g., selecting an appropriate number of lags) automatically. See, e.g., the GluonTS tutorial available at [C4] which does not use specific data preprocessing. We have furthermore contacted the authors of GluonTS who have confirmed that the methods do automatic preprocessing and are in principle designed to work out of the box. Similarly, also exponential smoothing and ARIMA need parameter adjustments (see also the response to R2 about citation [1]), which similarly is done automatically by the used ets and auto.arima functions, so that also here we use these functions with their default configurations, to enable a fair comparison.
> Thus, using default parameters seems the best option, as in general there is no easy fix to this problem and the line on how much preprocessing/tuning should be performed on the methods needs to be drawn somewhere. In fact, this criticism goes back all the way to the birth of the field of forecasting. In Makridakis et al. [C2] (published in 1979), in the discussion section, we see that those authors were accused that their unexpected results are a result of not having tuned the methods properly. Makridakis’ response to these comments was to organise the first M forecasting competition in 1982 [C3]. As such, the cleanest solution to the problem raised by the reviewer would be presumably to organise a forecasting competition. Though this is not possible as the data in our repository is already publicly available, as discussed under the responses to R2, we now provide a simple interface to users to enable integration of new forecasting models into our framework and evaluate them in the same way as we evaluate our baselines. This also enables comparing the performance of many novel forecasting models against our baselines.
>
> The data in the repository are raw data, except for missing value imputation as discussed in the Appendix.
>
> Regarding the reviewer’s statement of “statical methods tend to outperform neural networks in most datasets”. Across the main error measure used in the paper, mean MASE, excluding 2 datasets where NNs could not be run, statistical models have outperformed NNs on 20 datasets, Catboost has 3 wins, and NNs have 18 wins. Among the datasets where the statistical methods work well are datasets such as the M3 yearly/quarterly, and M4 yearly datasets that have very short, unrelated, noisy time series, on which consequently the simple Theta method performs best and even ETS and ARIMA are not able to outperform it. This is in line with decades of research in forecasting, where NNs, even with good regularisation and variance control, will unlikely be able to add value. On longer, more related and more regular datasets, NNs show in fact a very competitive behaviour overall in our experiments. Also responding to a comment from R2, we have added an according discussion to the paper (second last paragraph of Section 4.2).
>
> Regarding the comment of univariate forecasting, the reviewer is correct that we do not use covariates or other additional information in the paper. It is true that NNs could often benefit greatly from such additional information, as in the COVID deaths example mentioned by the reviewer. However, these additional predictors would be necessarily application specific, and it would be difficult to include all such different applications in a single repository. The task of autoregressive time series forecasting without external variables is nevertheless a common and important one, as highlighted by the citations to recent literature in the Introduction of the paper.
>
> We hope that our response, in particular around preprocessing that is already done automatically by GluonTS, provided clarifications, and we kindly request the reviewer to adjust their evaluation of our paper accordingly.
>
> [C1] Hewamalage et al. Recurrent neural networks for time series forecasting: current status and future directions. International Journal of Forecasting, 2021.
>
> [C2] S. Makridakis et al. Accuracy of forecasting: an empirical investigation. Journal of the Royal Statistical Society, 1979.
>
> [C3] Makridakis et al. The accuracy of extrapolation (time series) methods: results of a forecasting competition. Journal of Forecasting, 1982.
>
> [C4] https://ts.gluon.ai/tutorials/forecasting/quick_start_tutorial.html

---

### Official Review · Reviewer_xJBM · 2021-09-20
**Monash Time Series Forecasting Archive**

**Rating:** 6
**Confidence:** 4

**Strengths:**

1.	It collects and releases a collection of datasets from different domains. The attributes of time-series are very diverse, including various length, frequency, containing or not containing missing values, and univariate/multivariate. It will facilitate new design of time-series forecasting models in a comprehensive and complete evaluating protocol.
2.	It gives a comprehensive analysis of the dataset characteristics through feature extraction and PCA.
3.	It notifies the metrics problem of previous evaluation of time-series forecasting. To address these issues, the authors analyze the performance of baseline model with 10 error metrics and suggest that the selection of error metrics should be decided by specific applications.

**Weaknesses:**

1.	The paper only compares 13 baselines, while some recently published works are missed. It would be more informative to compare with more state-of-the-art models of time-series forecasting, such as Informer [1], StemGNN [2], and ST-GCN [3]. Moreover, as new time-series forecasting methods continuously emerge, it’s better to describe the unified evaluation protocol in more detail and provide an interface to implement new forecasting algorithms in a well-controlled setting.

2.	Time-series lack a unified metrics for evaluation. This paper raises this problem but leaves the decision to the owners of specific applications. To enable a fair comparison, at least the author should discuss the limitations of certain metrics and give suggestions about which metrics could be used as primary metrics for a certain kind of time-series.

[1] Zhou, Haoyi, et al. "Informer: Beyond efficient transformer for long sequence time-series forecasting." Proceedings of AAAI. 2021.

[2] Cao, Defu, et al. "Spectral Temporal Graph Neural Network for Multivariate Time-series Forecasting." Proceedings of the NeurIPS 2020.

[3] Yu, Bing, Haoteng Yin, and Zhanxing Zhu. "Spatio-Temporal Graph Convolutional Networks: A Deep Learning Framework for Traffic Forecasting." IJCAI. 2018.


**Additional Feedback:**

Refer to Weakness.

Figure 1 visualizes distinct density populations for various datasets. Therefore, it would be interesting to discuss the performance of baselines in terms of different density populations.

**Clarity:**

This paper is well written.


**Correctness:**

The dataset is constructed in a sound way. The experiments are performed reasonably.

**Documentation:**

The documentation is clear.

**Ethics:**

No concern.

**Relation To Prior Work:**

It has discussed the relation to prior works.

**Summary And Contributions:**

This paper presents the Monash Time Series Forecasting Archive, which contains 58 datasets (derived from 30 primary datasets, including 5 single very long time-series) across diverse domains and time-series attributes. Among them, 23 datasets are re-processed from publicly available datasets and 7 datasets are published by authors. In the experiments, the paper evaluates 13 different baseline forecasting models using a fixed origin evaluation scheme with 10 error metrics. Moreover, it gives a comprehensive analysis of the dataset characteristics through feature extraction and PCA. The effort of building this archive has a certain contribution to the research community of time series.

---

> ### Author Response · Authors · 2021-09-26
> **Responses to the Reviewer xJBM Comments**
>
> We thank the reviewer for highlighting the strengths of our paper and for giving valuable suggestions to improve it further.
>
> From the previous submission round of the reviewer suggestions and encouragement from the Area Chair, we had added 6 more baselines: CatBoost, Feed-Forward Neural Network (FFNN), DeepAR, N-BEATS, WaveNet and Transformer for our evaluations where all models except FFNN are recently published state-of-the-art machine learning and deep learning models. Executing all 13 methods across 43 datasets has been extensive, and it is also recognised by Reviewer 1 of the current round, as “a benchmark that presents results for a good amount of time series methods”.
>
> Nonetheless, we agree with the reviewer that this area of research is a fast moving one, and there are new methods available in the literature. Regarding the 3 papers cited by the reviewer:
>
> [1] This paper, that won a best paper award at AAAI 2021, is a very good example for why there is such a pressing need for a benchmark suite such as ours in the community. Those authors benchmark (among others) against not further specified versions of ARIMA, Prophet, and LSTM. They do this across 4 datasets, where the 2 public datasets out of those reflect electricity demand and weather data. As such, we can assume all data are highly seasonal; and long horizons are to be predicted. Only from Figure 9 in the Appendix (in the arxiv version of that paper), we can see that the ARIMA and LSTM benchmarks are quite unrealistic default versions of these algorithms. The ARIMA seems to be a non-seasonal ARIMA that quickly tapers off to a straight line. It completely fails to model the obvious and very stable seasonal pattern. A straightforward fix would be to use the DHR-ARIMA method that we use in our experiments. Similarly, also the LSTM fails to model the seasonality. This is presumably a lack of use of Fourier terms and/or long enough input and output windows in the LSTM. Thus, we are not confident that the experimental results of that paper are strong indications of that method’s merit (e.g., also on other types of series that are not highly seasonal), and inclusion of the method in our paper might stir controversy if the method does not perform well in our benchmarks. As such, we have opted for not including it, to not draw away the attention from the main contributions of our paper. Instead we have opted for slightly more established deep-learning techniques. However, a benchmark suite such as ours would help greatly to enable more rigorous evaluations of new methods in the future.
>
> [2] We agree with the reviewer that GNNs are a promising approach that may yield good results. As GNNs are multivariate methods, they would be applicable only to some of the datasets in our repository (see Table 1 in the paper). As we outline in the paper, we have currently excluded multivariate forecasting methods, to not further complicate our study that is already relatively extensive. However, we do agree that multivariate methods are a worthwhile subject of examination in the future, and the current benchmarks in the paper will be relevant for research into multivariate methods.
>
> [3] This work proposes a spatio-temporal graph network specifically for traffic forecasting. The model specifically considers the spatial attributes of traffic networks. The majority of our datasets do not belong to the transportation domain, and even for the datasets that belong to this domain, our repository doesn’t contain information on corresponding traffic networks, as we focus on general forecasting. Thus, the method proposed in [3] regretfully cannot be included in our study.
>
> Though we have discussed the reasons to not include the specific methods suggested by the reviewer, we strongly agree with the comments around continuous introductions of new forecasting models and how to keep our benchmarks up-to-date and relevant. Thus, as suggested, we now provide an interface for users to implement new forecasting models in a well-controlled setting. We have updated our github repository with more detailed instructions and code snippets explaining how to integrate new forecasting models to our framework. Accordingly, we have added a new paragraph in Section A.4 in the Appendix (last paragraph).
>
> We agree with the reviewer on the comment of providing recommendations on error metrics. Accordingly, we have now added a new paragraph into Section 4.2 (paragraph 2). We agree with the reviewer on the comment of discussing the performance of the baselines in terms of the feature analysis results. Accordingly, we have added a new paragraph (the second last paragraph) in Section 4.2 explaining some general relationships between the density populations and baseline results.
>
> We thank the reviewer for their valuable comments that have allowed us to improve the paper. We hope the reviewer can agree with our assessment that the paper has benefitted from those, and is now an even stronger contribution.

---

> > ### Comment · Reviewer_xJBM · 2021-09-29
> > **Thank you for the detailed explanation.**
> >
> > I agree with most of your comments. I think it's better to include Informer in the benchmark and add related discussion into the final version. Moreover, analyses in the multivariate time-series setting are encouraged in future works.

---

> > > ### Author Response · Authors · 2021-09-29
> > > **Thank you**
> > >
> > > We thank the reviewer for their response. Based on the reviewer suggestion, we have started running the Informer model with our datasets. The experiments are currently in progress and for the final version of the paper, we will include the results of the Informer model across all datasets in our repository as well as a related discussion.
> > >
> > > Furthermore, as suggested by the reviewer, we have now included a new paragraph at the end of Section 5 in the updated manuscript explaining the possible future work of this research which includes analysing the multivariate time series benchmarks.

---

### Official Review · Reviewer_ofkM · 2021-09-21
**A new repository with 25 datasets on multivariate time series plus 5 very long univariate time series**

**Rating:** 8
**Confidence:** 5
**Correctness:** No incorrectnesses were found.
**Clarity:** Very well written

**Strengths:**

- A new repository dedicated to time series, especially multivariate time series
- A new format designed to store time series
- A benchmark that presents results for a good amount of time series methods

**Weaknesses:**

- Only seven of the datasets were not yet public

**Additional Feedback:**



**Documentation:**

The supplementary file has all the necessary details. The data and the code were made public in the paper. I believe that more than having a maintenance plan, the goal is to be able to add new datasets to the repository which can be done easily with the new format to store time series.

**Ethics:**

No ethical issues.

**Relation To Prior Work:**

Only 23 of the 30 datasets were not previously publicly available. Anyhow, these 23 datasets were in different formats.

**Summary And Contributions:**

- a set of 25 multivariate time series datasets
- a new format to store time series
- Benchmark tests using a good amount of different methods

---

> ### Author Response · Authors · 2021-09-26
> **Thank you**
>
> We would like to thank the reviewer very much for the recognition of the importance of our work, and for providing such a high rating and such positive feedback.

---

### Author Response · Authors · 2021-09-26
**Updated Manuscript/Supplementary Materials**

We thank all reviewers for their valuable comments. We addressed them accordingly and updated the manuscript and supplementary materials.

---

### Decision · Program_Chairs · 2021-10-09

**Decision:**

Accept

**Comment:**

All the reviewers agree on acceptance. The authors are encouraged to use the reviewer feedback to improve the paper for its camera-ready version.